Effect of cirrhosis on prognosis in patients with acute-on-chronic liver failure: a systematic review and meta-analysis

Dong Chunyang 1
Liu Xiaoxiao 2
Zhang Dongshuai 1
Jiang Deyou 1 2264158135@qq.com
1 The First Affiliated Hospital, Heilongjiang University of Chinese Medicine , Harbin, Heilongjiang , China
2 The Second Affiliated Hospital, Heilongjiang University of Chinese Medicine , Harbin, Heilongjiang , China
Banerjee Priyanka
Electronic publication date: 2025 Sep 25
Publication date: 2025
Volume: 13
Electronic Location ID: e20049
Received 2025 Feb 26; Accepted 2025 Aug 15
Copyright: © 2025 Dong et al.
Copyright year: 2025
Copyright holder: Dong et al.
License: This is an open access article distributed under the terms of the Creative Commons Attribution License, which permits unrestricted use, distribution, reproduction and adaptation in any medium and for any purpose provided that it is properly attributed. For attribution, the original author(s), title, publication source (PeerJ) and either DOI or URL of the article must be cited.
License URL: https://creativecommons.org/licenses/by/4.0/

Keywords: Cirrhosis, Acute-on-chronic liver failure, Prognosis

Funding: The authors received no funding for this work.

==============================
Background

Acute-on-chronic liver failure (ACLF) is a prevalent complication among cirrhosis patients, whose high mortality is linked to cirrhosis combined with ACLF. Nevertheless, there is a paucity of systematic reviews. This study aimed to illustrate whether cirrhosis is a prognostic factor for ACLF.

Methodology

PubMed, Embase, and Cochrane Library were searched for observational studies that explored the connection between cirrhosis and ACLF prognosis from database inception to January 10, 2025. Pooled relative risk (RR) and 95% confidence interval (CI) were utilized for data analyses. Publication bias was estimated using Egger’s tests. The protocol was registered in the PROSPERO (CRD42025639557).

Results

This meta-analysis included 17 articles and 8,488 patients of ACLF under different diagnostic criteria. The analyses indicated that cirrhosis did not correlate with 28-day mortality of ACLF patients (RR = 1.08, 95% CI [0.84–1.39], p = 0.550, I2 = 88.8%) but independently predicted 90-day mortality (RR = 1.33, 95% CI [1.10–1.61], p = 0.004, I2 = 92.6%). Subgroup analyses of cirrhosis discovered no significant difference in 28-day and 90-day mortality between non-cirrhosis patients and those with compensated cirrhosis (p > 0.05). The 90-day mortality in decompensated cirrhosis patients was markedly higher than that in non-cirrhosis individuals (RR = 1.33, 95% CI [1.14–1.56], p < 0.001, I2 =64.2%).

Conclusions

Compensated cirrhosis did not correlate with the 28-day mortality of ACLF patients, while it was an independent risk factor for 90-day mortality.

Introduction

Acute-on-chronic liver failure (ACLF) is a severe clinical syndrome characterized by hepatic decompensation in addition to pre-existing chronic liver disease or cirrhosis, leading to organ failure and high rates of short-term mortality (Moreau et al., 2013). Currently, therapeutic options for ACLF include emergency treatment, organ support, and liver transplantation (Arroyo, Moreau & Jalan, 2020). Despite intensive care, short-term mortality within 90 days in ACLF patients is approximately 50% (Arroyo, Moreau & Jalan, 2020). Early diagnosis and accurate prognostic prediction have the potential to optimize ACLF management and improve survival (Zhang et al., 2023).

Several risk factors have been identified for the prognosis of ACLF. Cai et al. (2018) have argued the importance of systemic inflammatory responses in ACLF prognosis, in which inflammatory factors and markers, including white blood cells, lymphocytes, neutrophils, and platelets may be prognostic indicators of liver failure. Some articles have focused on the prognostic value of cirrhosis in ACLF. The World Gastroenterology Organization recommends that ACLF patients be classified into Type A chronic hepatitis, Type B compensated cirrhosis, and Type C decompensated cirrhosis (Jalan et al., 2014), with Type B and Type C being associated with cirrhosis. More than 25% of cirrhosis patients have ACLF-associated symptoms on admission (Hernaez et al., 2017). Recent studies have unveiled that cirrhosis can independently predict 3-month mortality in ACLF patients, and the mortality rate increases with ACLF grade (Shi et al., 2015; Zhao et al., 2018). A multi-national study from the Asia-Pacific region has indicated that 90-day mortality is markedly higher in Hepatitis B Virus (HBV)-ACLF patients in the cirrhotic group than the non-cirrhotic group, suggesting that cirrhosis may be a risk factor for ACLF and useful for risk stratification (Chen et al., 2019).

While previous findings suggest a connection between cirrhosis and ACLF, the influence of cirrhosis on ACLF mortality remains disputable. Many articles have reported that cirrhosis is not a risk factor for ACLF mortality and does not influence short-term mortality in individuals with liver failure (Li et al., 2017; Wang et al., 2014; Yang et al., 2012). Therefore, this systematic review and meta-analysis aimed to further explore whether cirrhosis is an independent risk factor for ACLF prognosis. This review is primarily intended for hepatologists, critical care clinicians, and translational researchers engaged in ACLF management, as well as public health policymakers involved in liver disease strategy development. By synthesizing conflicting evidence on cirrhosis-related risk stratification, our findings aim to equip clinicians with prognostic decision-making frameworks, empower researchers with prioritized knowledge gaps in cirrhosis-ACLF interactions, and enable policymakers to refine ACLF care pathways based on cirrhosis-specific mortality risks.

Materials and methodology

Retrieval strategies

This study followed the PRISMA guidelines (Page et al., 2021). The protocol was registered in the PROSPERO (CRD42025639557). PubMed, Embase, and Cochrane Library were thoroughly searched for relevant literature published in English until January 10, 2025. The searches were conducted by combining Medical Subject Headings (MeSH) terms and free words, including Liver Cirrhosis, Liver Fibrosis, Hepatic Fibrosis and Liver Failure, Hepatic Failure, Acute-On-Chronic Liver Failure, ACLF, and Mortality, Death*, and Survival. Additionally, references in relevant studies were also manually screened to determine potentially relevant articles. The detailed search strategies are presented in Table S1 of Supplemental Material.

Inclusion and exclusion criteria

Publications meeting the following criteria were enrolled: (1) adult patients with ACLF under different diagnostic criteria, which do not require cirrhosis as a prerequisite for diagnosis. (2) The case group included liver failure patients with cirrhosis, and the control group included liver failure patients without cirrhosis. (3) Outcome indicators included 28-day and 90-day mortality. (4) The type of study was observational study.

Exclusion criteria were as follows: (1) animal or cellular experiments, case reports, study protocols, reviews, editorials, letters, and conference articles; (2) studies with missing or grossly erroneous data; (3) duplicates; (4) studies without available full texts; and (5) studies incorporating duplicates of participants.

Data extraction

Two researchers (Dong and Liu) first screened the titles and abstracts of the included articles independently and then reviewed the full text to determine the final inclusion. Any discrepancies were tackled through discussion with a third researcher (Zhang). The following data were extracted from each eligible literature: author, year, country, study type, total sample size, age, sex, and the number of cirrhosis patients.

Quality evaluation

Two authors (Jiang and Dong) independently appraised the literature quality through the Newcastle Ottawa Scale (NOS) in the domains of selection, comparability, exposure, and outcome (Stang, 2010). A maximum of nine points can be awarded to the literature, and the study quality was rated as poor (<4 points), general (4–6 points), and good (≥7 points).

Statistical description

All data were analyzed via STATA 15.1. The relative risk (RR) and 95% confidence interval (CI) were employed to estimate the effect size of categorical variable outcomes. Heterogeneity across studies was evaluated using Cochrane P value and I2 statistics. p < 0.1 and I2 > 50% indicated pronounced heterogeneity, and then a random-effect model was applied; otherwise, a fixed-effects model was selected. When the heterogeneity was relatively high, subgroup analyses were performed on compensated and decompensated cirrhosis. Sensitivity analyses were subsequently conducted for all outcomes to determine the stability of the results. For the outcome with ≥10 articles, Egger’s funnel plot was leveraged to evaluate the publication bias (Sterne et al., 2011). p < 0.05 implied statistical significance.

Results

Literature screening results

A total of 12,026 articles were retrieved. After removing duplicates, we screened the titles and abstracts of the remaining 9,562 articles and selected 48 articles that met the criteria. Following reading the full texts, 17 articles were ultimately included (Chen et al., 2019; Han et al., 2015; Liu et al., 2021; Ma et al., 2023; Shi et al., 2015; Sun et al., 2021, 2009; Tang et al., 2021; Thanapirom et al., 2022; Wang et al., 2022; Wu et al., 2018; Xiao et al., 2024; 2021; Xu et al., 2021; Zhang et al., 2023, 2024; Zhou et al., 2024). Six articles did not report 28-day mortality and four articles did not report 90-day mortality. There were only 11 articles on 28-day mortality and 13 articles on 90-day mortality. Among the 17 articles, four studies compared outcome indicators between patients with compensated cirrhosis and those with decompensated cirrhosis (Shi et al., 2015; Tang et al., 2021; Xu et al., 2021; Zhang et al., 2024). The detailed flow chart for the screening process is delineated in Fig. 1.

Figure 1 Flowchart for literature screening.

Basic study information

After final screening, 8,488 patients were included, including 4,980 cases in the cirrhosis group and 3,508 in the non-cirrhosis group. Detailed characteristics are listed in Table 1. The included studies were all scored >6 points, indicating that all articles had high quality (Table 2). Among the 17 included studies, nine used the Asian Pacific Association for the Study of the Liver (APASL) criteria; six studies used the Chinese Group on the Study of Severe Hepatitis B and Chronic Liver Failure (COSSH-ACLF) criteria (Wu et al., 2018); one study used the Chinese Diagnostic and Treatment Guidelines for Liver Failure (2006 version); and one study used the European Association for the Study of the Liver-Chronic Liver Failure (EASL-AClF) criteria. It is important to note that under the EASL-ACIF criteria, cirrhosis is a prerequisite for ACLF (Moreau et al., 2013). Therefore, literature using EASL-ACIF as the diagnostic standard did not meet the inclusion criteria for this study. However, the literature included in this study, which used EASL criteria for diagnosis, explicitly stated that patients without cirrhosis were included (Liu et al., 2021), and therefore met the inclusion criteria for this study.

Table 1 Baseline characteristics applied for the association evaluation between cirrhosis and the outcome of liver failure (ACLF).

Author	Year	Country	Study type	Diagnostic criteria for ACLF	Total sample size	Age	Sex (Male/Female)	Cirrhosis number	Score	
Liu et al. (2021)	2021	China	Prospective study	EASL-ACLF*	157	43.70 ± 11.00	139/18	71	9	
Han et al. (2015)	2015	China	Prospective study	APASL	64	47.22 ± 11.34	46/18	59	9	
Wang et al. (2022)	2022	China	Retrospective study	APASL	1,177	45.06 ± 10.57	124/1,053	616	8	
Thanapirom et al. (2022)	2022	Thailand	Prospective study	APASL	1,621	44.06 ± 11.90	1,404/216	637	9	
Tang et al. (2021)	2021	China	Retrospective study	APASL	586	45.00 ± 12.60	503/83	391	8	
Chen et al. (2019)	2019	Korea	Retrospective study	APASL	709	45.66 ± 13.43	603/106	446	8	
Xiao et al. (2021)	2021	China	Prospective study	COSSH-ACLF	175	48.10 ± 11.90	119/56	175	9	
Wu et al. (2018)	2018	China	Prospective study	COSSH-ACLF	391	49.30 ± 11.00	335/56	299	8	
Shi et al. (2015)	2015	China	Retrospective study	APASL	540	48.80 ± 13.70	401/139	347	8	
Sun et al. (2021)	2021	China	Prospective study	APASL	290	45.05 ± 12.67	232/58	223	8	
Sun et al. (2009)	2009	China	retrospective study	CSS Guideline (2006)	204	46.80 ± 13.20	170/34	112	8	
Ma et al. (2023)	2023	China	Retrospective study	COSSH-ACLF	258	46.2 ± 11.70	221/37	202	8	
Xiao et al. (2024)	2024	China	Prospective study	APASL	207	44.7 ± 10.60	177/30	92	8	
Xu et al. (2021)	2021	China	Retrospective study	APASL	689	49 ± 11.90	545/144	254	8	
Zhang et al. (2023)	2023	China	Prospective study	COSSH-ACLF	357	46.65 ± 11.17	52/305	295	9	
Zhang et al. (2024)	2024	China	Prospective study	COSSH-ACLF	850	47.54 ± 11.21	702/148	635	8	
Zhou et al. (2024)	2024	China	Prospective study	COSSH-ACLF	213	48.60 ± 14.93	177/36	126	7	
Notes:

Abbreviation definitions: EASL-ACLF: European Association for the Study of the Liver-chronic Liver Failure; APASL: Asian Pacific Association for Study of the Liver; COSSH-ACLF: Chinese Group on the Study of Severe Hepatitis B of Liver-chronic Liver Failure; CSS Guideline (2006): Chinese Society of Hepatology, Diagnostic and treatment guidelines for liver failure (2006 version).

* Liver disease was chronic hepatitis (no cirrhosis) or compensated cirrhosis.

Table 2 Quality evaluation scores.

Study	Representation of the exposure cohort	Representation of the non-exposure cohort	Determination of exposure	None of the subjects reported a history of the researched disease at the start of the study	Comparability between the exposed and non-exposed cohorts	Method for result determination	Enough follow-up period or not	Integrity of follow-up	
Liu et al. (2021)	1	1	1	1	2	1	1	1	
Han et al. (2015)	1	1	1	1	2	1	1	1	
Wang et al. (2022)	1	1	1	0	2	1	1	1	
Thanapirom et al. (2022)	1	1	1	1	2	1	1	1	
Tang et al. (2021)	1	1	1	0	2	1	1	1	
Chen et al. (2019)	1	1	1	1	1	1	1	1	
Xiao et al. (2021)	1	1	1	1	2	1	1	1	
Wu et al. (2018)	1	1	1	1	1	1	1	1	
Shi et al. (2015)	1	1	1	0	2	1	1	1	
Sun et al. (2021)	1	1	1	1	1	1	1	1	
Sun et al. (2009)	1	1	1	0	2	1	1	1	
Ma et al. (2023)	1	1	1	0	2	1	1	1	
Xiao et al. (2024)	1	1	1	1	1	1	1	1	
Xu et al. (2021)	1	1	1	0	2	1	1	1	
Zhang et al. (2023)	1	1	1	1	2	1	1	1	
Zhang et al. (2024)	1	1	1	1	1	1	1	1	
Zhou et al. (2024)	1	1	1	1	1	1	0	1	

Comparison of relevant ACLF prognostic results

Impact of combined cirrhosis on mortality in ACLF patients

Eleven studies reported 28-day mortality of ACLF patients with comorbid cirrhosis, including 5,596 cases, with 3,409 in the cirrhosis group. The combined RR value was 1.08 (95% CI [0.84–1.39], p = 0.550, I2 = 88.8%), indicating no considerable difference in 28-day mortality between cirrhosis patients and non-cirrhosis individuals (Fig. 2). Nine studies reported 90-day mortality, involving 5,596 patients, with 4,366 cases in the cirrhosis group. The combined RR value using the random-effects model was 1.33 (95% CI [1.10–1.61], p < 0.001, I2 = 92.6%), indicating that 90-day mortality of ACLF in cirrhosis patients was 1.33 times higher than that in non-cirrhosis individuals (Fig. 3).

Figure 2 Impact of cirrhosis on 28-day mortality of ACLF patients (Shi et al., 2015; Han et al., 2015; Wu et al., 2018; Chen et al., 2019; Liu et al., 2021; Thanapirom et al., 2022; Tang et al., 2021; Xiao et al., 2021; Sun et al., 2021; Zhang et al., 2024; Zhou et al., 2024).

Figure 3 Impact of cirrhosis on 90-day mortality of ACLF patients (Sun et al., 2009; Shi et al., 2015; Wu et al., 2018; Chen et al., 2019; Liu et al., 2021; Thanapirom et al., 2022; Sun et al., 2021; Wang et al., 2022; Ma et al., 2023; Zhang et al., 2024; Xiao et al., 2024).

Impact of compensated cirrhosis and non-cirrhosis on mortality

Four articles were included in this analysis. The 28-day mortality was compared using a random-effects model, with a pooled RR of 1.26 (95% CI [0.85–1.88], p = 0.248, I2 = 84.6%), indicating no remarkable difference in 28-day mortality between compensated cirrhosis patients and non-cirrhosis individuals (Fig. 4A). Meanwhile, 90-day mortality was also compared, with a pooled RR of 1.38 (95% CI [1.01–1.88], p = 0.043, I2 = 77.7%), implying that 90-day mortality in compensated cirrhosis patients was 1.38 times higher than that in non-cirrhosis individuals (Fig. 4B).

Figure 4 Impact of compensated cirrhosis and non-cirrhosis on (A) 28-day mortality of ACLF patients and (B) 90-day mortality of ACLF patients (Shi et al., 2015; Wu et al., 2018; Liu et al., 2021; Tang et al., 2021; Zhang et al., 2024).

Impact of decompensated cirrhosis and non-cirrhosis on mortality

Four articles were included in this analysis. As for 28-day mortality, a random-effects model yielded a pooled RR value of 1.15 (95% CI [0.87–1.52], p = 0.327, I2 = 81.2%), without statistical difference in 28-day mortality in decompensated cirrhosis patients vs. non-cirrhosis individuals (Fig. 5A). Meanwhile, 90-day mortality was also compared via a random-effects model, yielding a pooled RR of 1.33 (95% CI [1.14–1.56], p < 0.001, I2 = 64.2%), implying that 90-day mortality in decompensated cirrhosis patients was statistically higher than that in non-cirrhosis individuals (Fig. 5B).

Figure 5 Impact of decompensated cirrhosis and non-cirrhosis on (A) 28-day mortality of ACLF patients and (B) 90-day mortality of ACLF patients (Shi et al., 2015; Wu et al., 2018; Tang et al., 2021; Zhang et al., 2024).

Impact of compensated and decompensated cirrhosis on mortality

Four articles compared 28-day mortality, and a fixed-effects analysis elicited a pooled RR of 1.31 (95% CI [1.16–1.49], p < 0.001, I2 = 24.2%), indicating that 28-day mortality in decompensated cirrhosis patients was 1.31 times higher than that in individuals with compensated cirrhosis (Fig. 6A). The random-effects analysis of 90-day mortality yielded a pooled RR of 1.27 (95% CI [1.16–1.39], p < 0.001, I2 = 0.0%), indicating that 90-day mortality in decompensated cirrhosis patients was 1.27 times higher than that in individuals with compensated cirrhosis (Fig. 6B).

Figure 6 Impact of compensated and decompensated cirrhosis on (A) 28-day mortality of ACLF patients and (B) 90-day mortality of ACLF patients (Shi et al., 2015; Xu et al., 2021; Tang et al., 2021; Zhang et al., 2024).

Sensitivity analysis and publication bias

Sensitivity analysis elicited that all results were relatively stable (Fig. 7). The funnel plot for 28-day mortality was symmetrically distributed (Fig. 8A) (p = 0.085), indicating no publication bias. The funnel plot for 90-day mortality showed asymmetry (Fig. 8B) (p < 0.05), indicating a potential publication bias.

Figure 7 Sensitivity analysis of (A) cirrhosis on 28-day mortality of ACLF patients (Liu et al., 2021; Thanapirom et al., 2022; Tang et al., 2021; Chen et al., 2019; Wu et al., 2018; Shi et al., 2015; Han et al., 2015; Xiao et al., 2021; Sun et al., 2021; Zhang et al., 2024; Zhou et al., 2024), (B) liver cirrhosis on 90-day mortality of ACLF patients (Liu et al., 2021; Thanapirom et al., 2022; Tang et al., 2021; Chen et al., 2019; Wu et al., 2018; Shi et al., 2015; Wang et al., 2022; Sun et al., 2021; Zhang et al., 2024; Ma et al., 2023; Xiao et al., 2024; Zhang et al., 2023), (C) decompensated and compensated cirrhosis on 28-day mortality of ACLF patients (Shi et al., 2015; Xu et al., 2021; Tang et al., 2021; Zhang et al., 2024), (D) decompensated and compensated cirrhosis on 90-day mortality of ACLF patients (Shi et al., 2015; Xu et al., 2021; Tang et al., 2021; Zhang et al., 2024).

Figure 8 Funnel plots of publication bias analysis on the impact of cirrhosis on (A) 28-day mortality and (B) 90-day mortality of ACLF patients.

Discussion

This is the first meta-analysis on the mortality of ACLF patients with comorbid cirrhosis. The current article analyzed recent data from 17 eligible studies. The findings indicated that cirrhosis was an important predictor of 90-day mortality, with no substantial difference in 28-day mortality. Similarly, subgroup analyses of cirrhosis (decompensated cirrhosis vs. non-cirrhosis; compensated cirrhosis vs. decompensated cirrhosis) uncovered that cirrhosis only served as an independent risk factor for 90-day mortality, rather than 28-day mortality.

Our findings were similar to previous studies. Liu et al. (2021) noted that the mortality rate of ACLF patients with comorbid cirrhosis was 2.89 times higher than that of non-cirrhosis individuals (p < 0.001). Several cohort studies have unveiled higher 90-day mortality rates in ACLF patients with sclerosis, ranging from 40% to 73.2% (Chen et al., 2019; Liu et al., 2021). These findings demonstrate the link between cirrhosis and high mortality in ACLF patients. The prognostic value of cirrhosis in 28-day short-term mortality in ACLF patients has also been reported. However, several articles have evidenced no link between cirrhosis and short-term survival in the ACLF population after adjustment for cirrhosis and its associated factors (Li et al., 2017; Wang et al., 2014; Yang et al., 2012).

ACLF has multiple predisposing factors. In addition to common infections, alcoholic hepatitis, and surgery, approximately 40% of ACLF patients have no clear predisposing factor (Bernal et al., 2015). Therefore, although we unveiled the prognostic ability of cirrhosis for 90-day mortality in ACLF patients, the underlying mechanisms still need further investigation. Existing studies have explained the mechanism from several aspects. First, in terms of immune function, cirrhosis patients typically exhibit systemic immune dysfunction, mainly manifesting as impaired monocyte function and decreased neutrophil chemotaxis, leading to a significant reduction in the body’s defense against infections (Rueschenbaum et al., 2020). Second, dysbiosis (dysregulation of the gut microbiota) in cirrhosis patients leads to increased bacterial translocation (Wei et al., 2013), which may exacerbate systemic inflammatory responses and trigger persistent endotoxemia and cytokine storms, thus contributing to organ dysfunction (Maloy & Powrie, 2011; Muñoz et al., 2019). Studies have evidenced that this dysregulation is more pronounced in individuals with decompensated cirrhosis and may be a major reason for their worse outcomes (Guo et al., 2021). Third, hemodynamic changes constitute another key mechanism. In cirrhosis patients, portal hypertension impairs intestinal barrier function, lowers body circulatory resistance, and increases cardiac output (Arab, Martin-Mateos & Shah, 2018). This hemodynamic disturbance causes inadequate renal perfusion and greatly increases the risk of acute kidney injury (Li et al., 2024). Persistent hemodynamic changes may lead to myocardial dysfunction and further exacerbate the instability of circulatory function (Shah et al., 2024). This is particularly pronounced in ACLF, creating a vicious cycle and resulting in unfavorable clinical outcomes in cirrhotic ACLF patients. Fourth, metabolic disturbances are also important in cirrhosis and affect ACLF prognosis. Cirrhosis patients have abnormal amino acid metabolism, especially lower levels of branched-chain amino acids (BCAAs) (Kawaguchi et al., 2011). BCAAs are critical for energy production and nitrogen metabolism, and deficiency in BCAAs impairs protein synthesis and complicates metabolic profiles, which is critical during acute decompensation (Davuluri et al., 2016; Kawaguchi et al., 2011). Furthermore, disturbed energy metabolism hinders tissue repair, especially during acute stress or shock (Wu et al., 2021). The liver is essential for the synthesis of coagulation factors, and its dysfunction can lead to coagulation disorders, particularly in ACLF, where the bleeding risk is greatly elevated due to impaired coagulation and underlying thrombocytopenia (Tian et al., 2022). Notably, glucose metabolism disorders have a particularly negative impact on cirrhosis patients. Early detection of impaired glucose tolerance is crucial for prognostic assessment of cirrhosis patients, as undiagnosed impaired glucose tolerance can lead to further complications and exacerbate systemic inflammatory responses (Nishida, 2017). These metabolic disturbances are particularly evident in response to acute shock and affect ACLF prognosis. These mechanisms do not work independently but form a complex pathophysiologic network. When cirrhosis progresses to the decompensated stage, the damage caused by the above mechanisms will become more severe, and the compensatory function of organs will be further undermined. Therefore, when such patients suffer acute shock, they are more prone to irreversible multi-organ failure and ultimately poor outcomes. Given no effective treatment for ACLF currently (Arroyo, Moreau & Jalan, 2020), understanding the interplay of these mechanisms is important for developing new therapeutic strategies and improving patient prognosis.

In this study, we conducted extensive searches to synthesize the most comprehensive evidence available on the prognostic value of comorbid cirrhosis in ACLF patients. However, this study has certain limitations. Firstly, most large-scale, multi-center studies are from China, which may introduce geographical and demographic biases. Secondly, liver failure can arise from multiple etiologies, including viral hepatitis, alcoholic hepatitis, cryptogenic cirrhosis, and drug-induced liver injury, each with distinct prognostic implications. Because most included studies are about HBV-ACLF, and the sample size for other etiologies is insufficient, this study cannot comprehensively analyze the prognosis of liver failure across different etiologies. Third, publication bias exists in some outcomes. Fourth, the study included ACLF patients diagnosed using different diagnostic criteria, but due to methodological limitations, it was not possible to perform subgroup analyses based on diagnostic criteria. Finally, there is heterogeneity, and the source is not identified. These limitations may impact the credibility of the results obtained. Future studies should include more research institutions from different geographical regions, including medical centers in Europe, North America, and other parts of Asia, to improve the representativeness and generalizability of the findings. Meanwhile, multicenter cohort studies specifically targeting different etiologies of liver failure can be designed to assess their prognostic characteristics and influencing factors. Such stratified studies will help to develop more targeted treatment strategies.

Conclusion

In conclusion, comorbid cirrhosis is an independent risk factor only for 90-day mortality of ACLF patients, rather than 28-day mortality. The 28-day mortality is more suitable for assessing the prognosis of patients with type A ACLF. Furthermore, the diagnostic criteria recommended by APASL should expand to decompensated cirrhosis, which is conducive to more standardized management of ACLF. Given some limitations in this study, large-sample multi-center studies are warranted to determine the impact of cirrhosis on ACLF prognosis.

Supplemental Information

Supplemental Information 1 PRISMA checklist.

Supplemental Information 2 Rationale.

Supplemental Information 3 Search strategy.

We would like to thank the researchers and study participants for their contributions.

Additional Information and Declarations

Competing Interests

The authors declare that they have no competing interests.

Author Contributions

Chunyang Dong conceived and designed the experiments, analyzed the data, authored or reviewed drafts of the article, and approved the final draft.

Xiaoxiao Liu conceived and designed the experiments, analyzed the data, prepared figures and/or tables, authored or reviewed drafts of the article, and approved the final draft.

Dongshuai Zhang performed the experiments, authored or reviewed drafts of the article, and approved the final draft.

Deyou Jiang conceived and designed the experiments, performed the experiments, prepared figures and/or tables, authored or reviewed drafts of the article, and approved the final draft.

Data Availability

The following information was supplied regarding data availability:

This is a Systematic Review/Meta Analysis.

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
