# Peer review of "Effect of cirrhosis on prognosis in patients with acute-on-chronic liver failure: a systematic review and meta-analysis"

_PeerJ, doi:10.7717/peerj.20049_

## Round 0.1 · original submission · Major Revisions

Reviewer 1 has raised serious concerns with the design and definitions used in your review. These must be addressed in full if the manuscript is to be considered further.

Reviewer 2 is concerned that the search protocol might have been registered retrospectively. The reference to Oct 2022 as temporal limit (in the Abstract only, as the methods section details January 2025 as limit) seems to be a typo, as the Forest plots in the results section contain data from works published in 2023 and 2024. Please make sure this discrepancy is resolved in your revision.

**Language Note:** The review process has identified that the English language must be improved. PeerJ can provide language editing services - please contact us at [email protected] for pricing (be sure to provide your manuscript number and title). Alternatively, you should make your own arrangements to improve the language quality and provide details in your response letter. – PeerJ Staff

Reviewer 1 ·

Basic reporting

This meta-analysis by Dong et al. aims to assess whether cirrhosis is an independent prognostic factor in patients with acute-on-chronic liver failure (ACLF). The authors pool data from 17 studies comprising 8,488 patients and report that cirrhosis does not significantly impact 28-day mortality but is associated with increased 90-day mortality. Subgroup analyses on compensated and decompensated cirrhosis attempt to refine this prognostic association further.

Experimental design

This paper, while clearly structured and methodologically well-presented, is fundamentally flawed in its conceptual design. According to the widely accepted EASL-CLIF definition, ACLF requires underlying cirrhosis as a precondition for diagnosis. The original landmark study by Moreau et al. (Gastroenterology, 2013) defined ACLF strictly in the context of acutely decompensated cirrhosis. Therefore, any attempt to compare outcomes between “ACLF patients with cirrhosis” versus “ACLF patients without cirrhosis” is conceptually invalid. The authors appear to conflate the broader APASL criteria—which permit ACLF diagnosis in patients with chronic liver disease without established cirrhosis—with the EASL definition, yet they do not make this crucial distinction explicit in their inclusion criteria, methodology, or interpretation of results.

Validity of the findings

Several included studies appear to define ACLF using HBV-related definitions (APASL) rather than cirrhosis-based definitions (EASL-CLIF), yet the authors do not address this heterogeneity adequately. As such, pooling these disparate populations (some with cirrhosis, some not) under the label of "ACLF" introduces significant conceptual and clinical confusion. It is difficult to interpret mortality outcomes meaningfully when the fundamental diagnosis itself lacks consistency across the dataset. Worse, the notion that patients without cirrhosis can be labeled as having “ACLF” in the EASL context contradicts the diagnosis's pathophysiological basis, which is rooted in acute decompensation of portal hypertension and systemic inflammation in cirrhosis.

Additional comments

In short, this study, despite its technical merit and statistical rigor, does not answer a clinically meaningful question—because it is comparing outcomes between populations that do not exist under the same diagnostic umbrella. If the intention was to explore outcomes in APASL-defined ACLF, this must be clearly stated in the title, abstract, methods, and discussion. As it stands, the manuscript promotes conceptual confusion and risks misinforming readers unfamiliar with the differences between competing definitions of ACLF.

The authors should be encouraged to:

Clarify whether this is an EASL-CLIF, APASL, or mixed-definition study.

Restrict analyses to one definition, or perform a subgroup meta-analysis stratified by diagnostic criteria.

Adjust the title to reflect the true population being studied.

Explicitly acknowledge that by EASL definition, cirrhosis is not a variable but a prerequisite.

Reviewer 2 ·

Basic reporting

Title & Abstract
Title:
The title is clear, informative, concise, suitable and reflective of the study. No further modification is required.
Abstract:
The abstract captured the main content and focus of the manuscript. It is structured in line with the journal and mentioned the aim, methods, results, and conclusions. However, the methods heading should be corrected to methodology to be in line with the journal instructions.
The abstract contains the essential information needed to understand objectives of the study, main methods, and significant findings of the study.
However, in the methods section, line 23, They said that they searched databases in October 2022. It is a too old and non-updated search. There are 32 months of non-appraised literature. They must update the search to 2025.
Lines 24-25, “The protocol was registered in the PROSPERO (CRD42025639557). This protocol was registered to PROSPERO in January 2025. Hence, this is a retrospective registration. Meta-analyses must be prospectively registered. PROPSERO itself do not allow retrospective registration.

Lines 31-32, the sentence “No considerable difference was noted in 28-day mortality between non-cirrhosis patients and those with decompensated cirrhosis (P>0.05)” carries the same meaning of the sentence just preceded it. So, delete this sentence as it is of repeated meaning.
Line 35, you stated that “Complicated cirrhosis did not correlate with the 28-day mortality of ACLF”. In the results section you did not mention complicated cirrhosis specifically. You mentioned cirrhosis in general. Conclusion must be supported with results. Hence, edit this conclusion to be in line with results.
The keywords are provided. The keyword correlation and meta-analysis are not suitable keywords. They are general terms.

Introduction
The introduction section is adequate and provides the essential information needed to understand magnitude of the problem, the importance, rationale, and objectives of the study. They highlighted the magnitude of ACLF and its burden. They referred to many factors that could affect prognosis and outcomes in such patients. They highlighted the scarcity of studies discussing the impact of cirrhosis on outcomes of ACLF and the urgent need for assessing such connection.

Figures & Tables
The paper contains 8 main figures, two main tables, and one supplementary table.
All tables and figures were cited in the text.

Experimental design

Material and Methods
They mentioned the methods in detail that enable its reproducibility. They adequately defined the search strategy, the searched databases, search terms, screenings, data extraction, quality assessment, and analysis.
It is a meta-analysis study that covers a sufficient time period from inception to 2025.
Line 77, you said you searched databases until January 10, 2025, while in the abstract you said until October 30, 2022. Please check and confirm.
It seems authors did not extract data that helped to assess your primary and secondary outcomes (mortality), then how did they calculate your outcomes?

Validity of the findings

Results
It is of great importance in the field of hepatology. This meta-analysis potentially helps clinicians and decision makers in predicting mortality in patients with ACLF. The results are adequately presented. The results are supported by the methods section and convey the main message of the study. They mentioned the searching databases, screening, quality assessment, and the results that confirm the impact of cirrhosis on ACLF. However, they started some sentences (lines 120, lines 130) with digital numbers. The sentences must begin with letter-based words. Hence, write these number in letter form, or add word/s before them. They provided a flowchart for the study.

Discussion
The discussion section is adequate. They discussed their main findings and objectives. The discussion section is supported by the results and methods sections. They adequately discussed their significant findings in comparison to the relevant published literature. They highlighted the impact of cirrhosis on ACLF and discussed factors affecting outcomes in such patients. They thoroughly discussed the proposed mechanisms of impact of cirrhosis on ACLF and supported their theories with relevant articles.
Lines 179-182, the sentences “The findings indicated that cirrhosis was an important predictor of 90-day mortality, with no substantial difference in 28-day mortality. Similarly, subgroup analyses of cirrhosis uncovered that cirrhosis only served as an independent risk factor for 90-day mortality, rather than 28-day mortality” these two sentences carry the same meaning. Predictors of mortality equal risk factor. What is the statistical test used for each one of these sentences? Line 230, this paper, correct paper to study. Write “this study”. Limitations of the study were mentioned, and they are adequate. They mentioned their limitation including heterogeneity of included studies, variant etiologies of cirrhosis, and the small sample size of some patients’ subgroups. Recommendations for further large multicenter prospective studies were mentioned.

Conclusion
This section is adequately written. They concluded their significant findings regarding impact of cirrhosis on outcomes in patients with ACLF and the main factors affecting mortality.

---

## Round 0.2 · accepted · Accept

Thank you addressing the reviewer's comments and submitting the revised manuscript.

Reviewer 1 ·

Basic reporting

The authors have now adequately clarified the diagnostic criteria used across included studies, appropriately acknowledged the definitional heterogeneity in ACLF, and revised the manuscript to address conceptual concerns; I recommend the manuscript be accepted in its current form.

Experimental design

-

Validity of the findings

-